# TRPV3 Ion Channel: From Gene to Pharmacology

**DOI:** 10.3390/ijms24108601

**Published:** 2023-05-11

**Authors:** Aleksandr P. Kalinovskii, Lyubov L. Utkina, Yuliya V. Korolkova, Yaroslav A. Andreev

**Affiliations:** 1Department of Molecular Neurobiology, Shemyakin-Ovchinnikov Institute of Bioorganic Chemistry, Russian Academy of Sciences (IBCh RAS), 16/10 Miklukho-Maklay Str., 117997 Moscow, Russia; kalinovskii.ap@gmail.com (A.P.K.); july@mx.ibch.ru (Y.V.K.); 2Institute of Molecular Medicine, Sechenov First Moscow State Medical University, Trbetskaya Str. 8, Bld. 2, 119991 Moscow, Russia; lyuba_utk@mail.ru

**Keywords:** TRPV3, ligands, itch, dermatitis, hair growth, skin regeneration, pain, normal and pathological conditions, therapeutics

## Abstract

Transient receptor potential vanilloid subtype 3 (TRPV3) is an ion channel with a sensory function that is most abundantly expressed in keratinocytes and peripheral neurons. TRPV3 plays a role in Ca^2+^ homeostasis due to non-selective ionic conductivity and participates in signaling pathways associated with itch, dermatitis, hair growth, and skin regeneration. TRPV3 is a marker of pathological dysfunctions, and its expression is increased in conditions of injury and inflammation. There are also pathogenic mutant forms of the channel associated with genetic diseases. TRPV3 is considered as a potential therapeutic target of pain and itch, but there is a rather limited range of natural and synthetic ligands for this channel, most of which do not have high affinity and selectivity. In this review, we discuss the progress in the understanding of the evolution, structure, and pharmacology of TRPV3 in the context of the channel’s function in normal and pathological states.

## 1. Introduction

Transient receptor potential (TRP) channels are a superfamily of cation-permeable channels that respond to various extracellular and intracellular stimuli and are involved in a number of physiological processes and pathological conditions. The TRP channel superfamily consists of seven subfamilies based on sequence homology: TRPC (canonical), TRPV (vanilloid), TRPM (melastatin), TRPP (polycystin), TRPML (mucolipin), TRPA (ankyrin), and TRPN (no mechanoreceptor potential C-like, or NOMPC-like) [1,2]. The TRPV subfamily includes TRPV1-TRPV6 members, which are widely expressed in both non-sensory and sensory cells, have high sequence similarity in different species, and display specific activation mechanisms and physiological functions (latest reviews [3,4,5]).

TRPV3 was discovered in 2002 when Peier et al. cloned the TRPV3 cDNA from the skin of newborn mice [6]. Independently of this group, TRPV3 was also found in human cells: Xu et al. discovered a cDNA encoding TRPV3 by analyzing a human brain cDNA library [7], and Smith et al. cloned the TRPV3 gene [8]. To date, TRPV3 is, relatively, an insufficiently studied non-selective cation channel that belongs to thermosensitive ion channels and that acts as a sensor of innocuous heat. It has been established that TRPV3, which is abundantly expressed in keratinocytes, is involved in the maintenance and functioning of the skin barrier and mediates the development of inflammatory skin diseases, wound healing, the transmission of pain signals, and hair morphogenesis. Dysfunctional mutations in the *trpv3* gene cause the genetic Olmsted syndrome, characterized by palmoplantar keratoderma with periorificial keratotic plaques, inflammation, and severe itching [9]. There are number of reports of natural and synthetic ligands of TRPV3 that modulate its function [9,10] but the lack of potent and highly selective pharmacological tools still deters studying TRPV3 on the molecular level. In this review, we provide up-to-date information about TRPV3 at the levels of gene and protein, and we analyse the usage of the known ligands as molecular tools for the study of TRPV3 in the context of normal and pathological conditions, as well as their potential as experimental therapeutics.

## 2. Molecular Characteristics of TRPV3

### 2.1. Gene and Evolution

The human *trpv3* gene is located on chromosome 17p13.2 on the antisense strand of DNA. The length of the entire gene is more than 47 kb and includes 18 exons, of which 17 are encoding and the 18th is an upstream non-coding exon [7,8]. The gene is adjacent to the gene of another family member, *trpv1*, on chromosome 17, with a distance between them of 7.45 kb, and both genes are located in the same transcriptional orientation. Such a close arrangement of the *trpv1* and *trpv3* genes is typical for animals with the *trpv3* gene in their genomes [11,12,13]. Only in the platypus genome was the *trpv7* gene found on the sense DNA strand between the *trpv3* and *trpv1* genes [12].

*Trpv3* homologs are found in such vertebrates as amphibians, reptiles, birds, and mammals, with more than 130 orthologs in total [14]. *Trpv3* gene was not found in fishes. Saito et al. suggested that TRPV3 arose in a common ancestor of fish and tetrapods, and was then was lost in the process of the evolution of fishes [11]. Morini et al. supported an early origin of TRPV3, as the gene was present in four cartilaginous fishes (elephant shark, spotted catshark, whale shark, and thorny skate) and sarcopterygians (from coelacanth to human) but is absent in actinopterygians. In the *Latimeria chalumnae* genome, there are three *trpv3* genes that are grouped in a phylogenetic tree, indicating a duplication [15]. Saito et al. also supposed that the common ancestor had three *trpv* genes organized in tandem, *trpv2-trpv1-trpv3*, and several genes were later inserted between *trpv2* and *trpv1*, while the region of the chromosome containing *trpv1* and *trpv3* remained conserved [11].

Cetaceans, manatees, and hippopotamus have a unique epidermal structure: a thick stratum spinosum, no stratum granulosum, and a parakeratotic stratum corneum to adapt to the environment. Concurrently, *trpv3* gene was inactivated in 18 species of whales and dolphins due to premature stop codons, initial codon mutations, and splice site mutations. The *Trpv3* gene was under relaxed selective pressure in cetacean lineages with both intact and inactivated *trpv3*, and in manatees and hippopotamus. *Trpv3* knockout mice have a thickened stratum spinosum and defective stratum granulosum and stratum corneum [16], which is consistent with the cetacean phenotype. This assumes that *trpv3* degradation is a genomic trace of epidermal development in aquatic and semiaquatic mammals [17].

In mammoth, *trpv3* genes had a mutation leading to N647D substitution in a well-conserved site in the outer pore loop, and this mutation is thought to be positively selected. A mutation at site 647 affects temperature-dependent gating of the channel [18], and reconstructed mammoth TRPV3s were activated at a lower temperature (~29 °C), so this gene may have contributed to the evolution of cold tolerance in mammoths [19].

In western clawed frogs, the central part of TRPV3 is highly homologous to other tetrapod species, but N- and C-terminal regions are divergent from them. In the genome of the western clawed frog, there are no homologous sequences for the N- and C-terminus of mammalians. It is interesting that the western clawed frog TRPV3 is not stimulated by heat, but rather by cool temperature (~16 °C) induced large currents in oocytes expressing this protein. The optimal temperature for this species is 22–28 °C, and the values below 18–20 °C are injurious. So, it can be postulated that the TRPV3 channel detects noxious low temperatures in the western clawed frog [12].

Price et al. identified that the *trpv1/trpv3* intergenic region is enriched with human-specific SINE-VNTR-Alu (SVA) retrotransposon insertions [20]. The SVA, located approximately 400 bp downstream the *trpv1 3′UTR* and 5.7 kb upstream of the *5′ trpv3* transcriptional start site, serves as a cis-regulatory element for the *trpv3* gene. The deletion of all SVA alleles resulted in a significant decrease in TRPV3 mRNA expression, while the effects of the SVA deletion on TRPV1 mRNA expression could not be determined. In heterozygous ΔSVA clones, the general trend was to increase the variability in mRNA expression of *trpv1* and *trpv3* genes. In contrast to homozygous ΔSVA clones, there was no correlation between gene expression values in heterozygous ΔSVA clones. This indicates that the loss of SVA in the intergenic region between both genes may disrupt the regulatory mechanisms that regulate co-expression in native cells.

### 2.2. Protein

Mature mouse TRPV3 is a four-fold symmetrical tetramer, and each subunit consists of 790, 791 or 765 amino acids, depending on the splicing variant. Each subunit has a transmembrane domain (TMD) that includes six transmembrane α-helical segments (S1–S6). The first four transmembrane segments form a bundle and comprise the S1–S4 domain, also known as the voltage-sensor-like (VSL) domain, and S5 and S6 with a pore loop (P-loop) form a pore domain. Massive cytoplasmic N- and C-termini form a so-called intracellular skirt. The N-terminus contains six ankyrin repeats, which form an ankyrin repeat domain (ARD) followed by the ARD-TMD linker domain that includes a β-hairpin (composed of β-strands, β1 and β2) and a helix-turn-helix motif (composed of linker helices, LH1 and LH2) [21]. As shown by the crystal structure of the ARD of mouse TRPV3, each ankyrin repeat is typically a 33 amino acid residue motif that forms two antiparallel α-helices separated by a turn, followed by a loop, connecting neighbouring repeats. TRPV3-ARD contains a long loop between repeats 3 and 4 (finger 3), which is not typical for other members of the TRP family. It curves towards finger 2 and is stabilized by interaction with the inner helices of repeat 3 and repeat 4. According to the surface electrostatic potential of TRPV3-ARD, there is a positively charged region for the interaction with the triphosphate group of adenosine triphosphate (ATP), but the hydrophobic groove for binding the adenine base of ATP is closed by bent finger 3, creating a potential steric collision between finger 3 and a ligand such as ATP. Binding of calmodulin to ARD can induce conformational changes in finger 3, resulting in inhibition of TRPV3 function. It can be assumed that TRPV3-ARD finger 3 can function as a switch in TRPV3 regulation upon ligand binding [22]. The C-termini has an amphipathic TRP helix that is parallel to the TMD, after which TRPV3 forms a C-terminal hook that ends with a β-strand (β3). The C-terminal hook, consisting of 19 residues, is unique for TRPV3 and is not found in other TRP channels. The β3 connects to the β-hairpin in the linker domain via hydrogen bonds to form a three-stranded β-sheet that, together with the C-terminal hook, participates in intersubunit interactions with the ARD that stabilize the elements of the intracellular skirt together (Figure 1) [6,7,8,23,24,25].

Smith et al. suggested that TRPV1 and TRPV3 can form heteromeric channel assemblies, as they are co-localized and co-expressed in sensory neurons [8]. Hellwig et al. tested this hypothesis by fluorescence resonance energy transfer (FRET), co-localization, and co-immunoprecipitation analysis, and they did not reveal a significant interaction between TRPV1 and TRPV3, i.e., these proteins preferentially formed homomeric complexes in cultured cells [26]. Later, it was shown that TRPV1 and TRPV3 can randomly combine into tetramers, and the composition of the tetramers can be different: 3 TRPV1 + 1 TRPV3; 2 TRPV1 + 2 TRPV3; 1 TRPV1 + 3 TRPV3. In this case, the properties of heteromeric channels depend on how many subunits of which type are included in its composition [27]. Later, Cheng et al. constructed heteromeric channels with two TRPV1 subunits and two TRPV3 subunits and proved the possibility of the presence and functioning of such channels in cells [28]. The formation of heteromeric ion channels is not too surprising, as channels of this type were revealed for the TRPC and TRPM subfamilies [28,29,30,31]. One of the main conditions for assembling different molecules is a high degree of their sequence homology [32]. Although TRPV3 sequence has a rather low homology compared to other members of the TRPV family (about 42%, 43%, 41%, and 28% of similarity with TRPV1, TRPV2, TRPV4, and TRPV5/6, respectively [8]), the ability of TRPV3 molecules to form heteromeric tetra complexes cannot be ruled out.

TRPV3 is a Ca^2+^-permeable nonselective cation channel [7]. For ruminants, rumen-expressed TRPV3 channels are important not only for maintaining Ca^2+^ homeostasis but also for the transport of NH_4_^+^, Na^+^, and K^+^ across the rumen [33,34,35,36]. The expression of human TRPV3 in cells also stimulates the influx of NH_4_^+^ and may participate in nitrogen metabolism [37].

The TRPV3 ion channel can be gated by both thermal and chemical stimuli. Generally, TRPV3 is gated by innocuous heat, and the temperature threshold of mouse TRPV3 was calculated as 31 °C [38]. Repeated activations of both native or heterologously expressed TRPV3 lead to the channel sensitization, a property that is unique among other TRP-channels that show time-dependent desensitization upon a prolonged stimulation [7,8,39,40,41]. The initial temperature-dependent activation of TRPV3 can require a rather high temperature (>50 °C), and, subsequently, TRPV3 activates at lower temperatures (>33 °C) [42]. The sensitization of mouse TRPV3 showed calcium dependence and was mediated by calmodulin acting at the N-termini and by an D641 residue at the pore loop [41]. Structurally, the sensitized state is an intermediate state between the closed and the open conformations. It is highly temperature sensitive, and is accompanied by the withdrawal of the vanilloid-site lipid as well as massive changes in the secondary structure and positions of S2–S3 loop and the N- and C-termini. The heat stimulation induces the so-called conformational wave, i.e., mutually-dependent structural changes that propagate from one domain to another. This involves the vanilloid site, S2–S3, and S4–S5 linkers, the pore domain, TRP helix, linker domain, three-stranded β-sheet, loops of the ARD, and N- and C-termini. Cooperativity of different domains explains why numerous amino acid substitutions over the entire channel alter thermal sensitivity and complicate the identification of a specific domain that assumes the role of a temperature sensor or a conformational wave trigger [43].

## 3. Ligands of TRPV3

### 3.1. Natural Agonists

A variety of natural compounds exhibit TRPV3 agonism (Table 1). Notably, most of them belong to terpenoids, natural products derived from C5 isoprene units. A group of monoterpene TRPV3 agonists includes components of essential oils, skin sensitizers, and allergens, e.g., thymol, carvacrol, camphor, etc. Structure-activity relationships within the monoterpenes were addressed in a study [44] that revealed the importance of a secondary hydroxyl group for effective channel activation.

A sesquiterpene farnesyl pyrophosphate (FPP) was identified as the first endogenous molecule that potently (in nanomolar concentrations) activated TRPV3 without any substantial activation of other sensory TRP channels. Chemically related compounds like farnesol (alcoholic form of FPP), geranyl, and geranylgeranyl pyrophosphates did not activate TRPV3 at the micromolar range [45], but isopentenyl pyrophosphate showed a potent antagonistic activity [46].

Incensole acetate, a diterpene of *Boswellia* resin, or frankincense, which has been used in religious rituals since ancient times, exerted selective and potent agonism toward TRPV3 [47]. The structure-activity study of a series of diterpenoids revealed that the macrocyclic cembrane skeleton is a hot-spot structure for targeting TRPV3 with a good pharmacological potential. Of special relevance was serratol, a component of Indian frankincense, that was identified as a submicromolar agonist, which was more potent than incensole acetate [48]. Dimer ent-labdane diterpenoids from a medicinal plant, *Andrographis paniculata*, showed the ability to activate TRPV1–4 channels. Bisandrographolide B showed the tightest binding to TRPV3, but it also bound TRPV1 with a greater affinity [49].

Potent TRPV3 activators have been found among cannabinoids, a group of terpenophenolics found in *Cannabis sativa*. They are best known as ligands of cannabinoid receptors, CB1 and CB2, but at least six subtypes of TRP-channels (TRPV1-V4, TRPA1, TRPM8) are also known as their additional targets [50]. Cannabidiol and Δ^9^-tetrahydrocannabivarin produced potent agonism of rat TRPV3, while other tested compounds were less efficacious but could potently desensitize the channel to further stimulation by carvacrol [51].

**Table 1 ijms-24-08601-t001:** Agonists of TRPV3 ion channel.

Compound	Potency		Confirmed by	Refs.
	Species	EC_50_, µM		
Diphenyl-containing compounds				
2-Aminoethoxydiphenyl borate	Mm	28.3–165.8	e/p *	[32,52]
	Hs	78	c.i. **	[53]
Diphenylboronic anhydride	Mm	64.1–85.1	c.i., e/p	[52]
Drofenine	Hs	207	c.i., e/p	[53]
Other synthetic compounds				
KS0365	Mm	5.08	c.i., e/p	[54]
Monoterpenes				
6-tert-butyl-m-cresol	Mm	370	e/p	[44]
Carvacrol	Mm	490	e/p	[44]
	Hs	438	c.i.	[53]
Thymol	Mm	860	e/p	[44]
Citral	Mm	926	e/p	[55]
Dihydrocarveol	Mm	2570	e/p	[44]
(−)-Carveol	Mm	3030	e/p	[44]
(+)-Borneol	Mm	3450	e/p	[44]
Camphor	Mm	6030	e/p	[44]
(−)-Menthol	Mm	20,000	c.i., e/p	[56]
Sesquiterpenes				
Farnesyl pyrophosphate	Hs	0.1311	c.i., e/p	[45]
Diterpenes				
Serratol	Rn	0.17	c.i.	[48]
Incensole acetate	Mm	16	c.i., e/p	[47]
Bisandrographolide B	Mm	40.5	MST ***, e/p	[49]
Cannabinoids				
cannabidiol	Rn	3.7	c.i.	[51]
Δ^9^-tetrahydrocannabivarin	Rn	3.8	c.i.	[51]

* e/p—electrophysiology, ** c.i.—calcium imaging, *** MST—microscale thermophoresis.

### 3.2. Synthetic Agonists

An important group of TRPV3 activators is diphenyl-containing compounds, with a special place for 2-aminoethoxydiphenyl borate (2-APB). 2-APB was the first discovered non-thermal activator of TRPV3 that produced a robust activation and sensitization to heat of recombinant channels and native channels in mouse keratinocytes [32]. 2-APB also acts on multiple cell-surface canonical, melastatin- and vanilloid-subtype TRP channels [57]. Since its introduction, 2-APB has been a valuable tool for the analysis of TRPV3 physiology, structure, and the identification of antagonists.

Diphenyl-containing compounds, diphenylboronic anhydride (DPBA), and diphenyltetrahydrofuran (DPTHF), which are structurally related to 2-APB, also modulated the channel: DPBA acted as a TRPV3 agonist, whereas DPTHF exhibited prominent antagonistic activity [52]. An antispasmodic agent drofenine-HCl showed selective agonism to TRPV3 with comparable potency to 2-APB and no effect on TRPA1, TRPM8, TRPV1, TRPV2, and TRPV4 in submillimolar concentrations. The chemical structure of drofenine is similar to 2-APB, and the two compounds likely have similar but not identical binding sites that both contain H426. Drofenine also caused cytotoxicity, exhibiting greater potency than 2-APB and carvacrol [53].

A novel synthetic activator of TRPV3, KS0365 showed a higher efficacy and potency than 2-APB. The compound also acted non-selectively, acting on TRPV1 and TRPV2 channels and triggering intracellular calcium release at higher concentrations [54].

### 3.3. Natural Antagonists

Isopentenyl pyrophosphate (IPP), an upstream metabolite of farnesyl pyrophosphate (FPP) that was previously introduced as a TRPV3 activator, showed the opposite, inhibitory, effect on the channel (Table 2). The compound acted in nanomolar concentrations, but, in a row of other sensory TRP channels (TRPV1-V4, TRPA1, TRPM8), additionally potently inhibited TRPA1 [46]. An endogenous lipid, 17(R)-resolvin D1, was shown to suppress TRPV3-mediated activity at nanomolar and micromolar concentrations [58].

Medicinal plants are a valuable source of TRPV3 antagonists that vary in potency, selectivity, and chemical nature. Effective TRPV3 inhibition was found for coumarin osthole from a medicinal plant *Cnidium monnieri* [59]; phenylethanoid glycoside forsythoside B, isolated as an active ingredient of a herb *Lamiophlomis rotate* [60]; citrusinine II, an acridone alkaloid from plant *Atalantia monophylla* [61]; isochlorogenic acid A, and isochlorogenic acid B, two dicaffeoylquinic acid isomers from a herb *Achillea alpine* [62]; and scutellarein, one of major flavonoids of *Scutellaria baicalensis* Georgi [63]. The selectivity profile of plant inhibitors varies, for example, and a variety of molecular targets have been reported to interact with osthole, including ion channels, e.g., TRPV1 [64], TRPA1 [65], CFTR chloride channel [66], and voltage-gated sodium channels [67].

An array of natural guanidine alkaloids showed a non-selective inhibitory activity toward TRPV3 [68,69,70,71]. The most potent action was detected for monanchomycalin B, an alkaloid with a pentacyclic core and a spermidine moiety, isolated from a marine sponge *Monanchora pulchra*. The compound inhibited rTRPV1, mTRPV2, and hTRPV3, but had no activity to rTRPA1 [68]. Echinochrome A, a naphthoquinoid pigment from sea urchins, inhibited TRPV3 and Orai1 channels in a low micromolar range and modulated two-pore K+ channels [72].

### 3.4. Synthetic Antagonists

Ruthenium red, a cationic dye, is a non-selective pore blocker of TRPV- and multiple other channels, and it is sometimes used as a reference drug in TRPV3 studies [25,73].

**Table 2 ijms-24-08601-t002:** Antagonists of TRPV3 ion channel.

Compound	Potency		Confirmed by	Refs.
	Species	IC_50_, µM		
Endogenous compounds				
Isopentenyl pyrophosphate	Hs	7.5	c.i. *, e/p **	[46]
17(R)-resolvin D1	Hs	0.398	c.i., e/p	[58]
Components of medicinal plants				
Osthole	Hs	37	c.i., e/p	[59]
	Mm	20.5	c.i., e/p	[74]
Forsythoside B	Hs	6.7	c.i., e/p	[75]
Verbascoside	Hs	14.1	e/p	[76]
Citrusinine-II	Mm	12.43	c.i., e/p	[61]
Isochlorogenic acid A	Hs	2.7	c.i., e/p	[62,77]
Isochlorogenic acid B	Hs	0.9	c.i., e/p	[62,77]
Scutellarein	Mm	1.18	e/p	[63]
Marine products				
Monanchomycalin B	Hs	3.25	c.i.	[68]
Echinochrome A	Hs	2.11	e/p	[72]
Clinical drugs				
Dyclonine	Mm	3.2	e/p	[78]
Bupivacaine	Hs	170	e/p	[79]
Ropivacaine	Hs	280	e/p	[79]
Other synthetic compounds				
7c	Hs	1.05	e/p	[80]
8c	Hs	0.086	e/p	[80]
74a	Hs	0.38	c.i., e/p	[81]
Ruthenium red	Mm	-	c.i., e/p	[6]
Diphenyltetrahydrofuran	Mm	Biphasic: 6, 151.5 (−80 mV); 10, 226.7 (+80 mV)	c.i., e/p	[52]
26E01	Mm	8.6	c.i., e/p	[82]
PC5	Mm	2.63	e/p	[83]
Trpvicin	Hs	0.41	e/p	[84]

* c.i.—calcium imaging, ** e/p—electrophysiology.

A local anaesthetic, dyclonine, potently and selectively inhibited TRPV3 currents. Some other local anaesthetics also modulated TRPV3 activity. Lidocaine and its analogs suppressed 2-APB-induced currents in TRPV3-expressing *Xenopus laevis* oocytes in low and submillimolar concentrations, which are, however, pharmacologically relevant for local anaesthesia. Weak activating properties of drugs were also detected in high concentrations. The effects are likely not specific among other TRP-channels since, in previous reports, different modulation of TRP-channels by lidocaine and QX-314 was shown [79].

Two cinnamate ester derivatives, 7c and 8c, were identified as potent TRPV3 antagonists that showed no off-target activity to TRPV1 and TRPV4 [80]. A novel compound 26E01 effectively blocked activation of mouse and human TRPV3 by heat and 2-APB, and did not exert any effect on TRPV1, TRPV2, or TRPV4. The compound also blocked native TRPV3 channels in the mouse keratinocyte cell line Kera-308, colonic epithelial cell line DLD-1, and primary colonic crypts isolated from mouse distal colon [82]. A compound named PC5 was reported as a TRPV3 inhibitor, whose structure was optimized by computational methods from a hit compound found in chemical libraries. PC5 completely suppressed mTRPV3 currents in low micromolar concentrations, but also acted on rTRPV1, mTRPC6, and rTRPM8, albeit not showing a complete block [83]. The compound 74a was reported to potently block TRPV3 and possess an optimized pharmacological profile. The evaluation of selectivity conducted on an extensive board of receptors and ion channels, including those involved in pain perception, showed no significant binding to adverse targets [81]. Trpvicin was identified as another potent and subtype-selective inhibitor of TRPV3 [84].

## 4. Structural Interactions of TRPV3 with the Ligands

Critical advances in the understanding of how pharmacological ligands interact with the TRPV3 ion channel on the structural level have become possible due to high resolution cryogenic electron microscopy (cryo-EM). Structural biology data are normally supported and confirmed by site-directed mutagenesis coupled with functional assays and molecular simulations. Characterization of binding sites and mechanisms of action provides valuable data for the rational design of new molecules for target-directed therapeutics (Figure 2).

Structural insights into the gating of mTRPV3 by the agonist 2-APB have been obtained [21]. The agonist binds three allosteric sites: the top region of S1–S3 transmembrane domain, the base of S1–S4, and the ankyrin repeat domain—transmembrane domain (ARD-TMD) linker site. According to the proposed mechanism, when binding to the top S1–S4 site, 2-APB displaces the S1–S2 loop, which resides there in the apo-state and expands the top S1–S4 bundle. The cleft between S4 and S6 narrows, squeezing out a lipid from the vanilloid site, and the S1–S4 and pore domain interface rearranges in a manner that supports channel opening. The additional two sites do not exhibit strong gating-associated conformational changes, but they are likely necessary for the stability of the pore, S1–S4, and skirt domains during the channel opening [21,25]. In the study on human TRPV3, the ARD-TMD linker site was identified as the only site that bound 2-APB, suggesting that cytoplasmic interface guides the channel gating in this case [85]. Molecular properties of the interaction of carvacrol with hTRPV3 were obtained by molecular docking and site-directed mutagenesis. A unique interaction site was identified at the S2–S3 linker, with Leu508 being the most critical residue for the channel activation [86].

Structural data for the TRPV3 inhibition was obtained for osthole. The cryo-EM structure of mTRPV3 in complex with an antagonist revealed that osthole outcompetes 2-APB in the agonist’s binding sites in the base S1–S4 and the ARD-TMD linker domain sites and converts the channel pore into a state with a widely open selectivity filter and a closed intracellular gate [74]. As also shown by cryo-EM and confirmed by site-directed mutagenesis, dyclonine binds to the sites at S6 inside the mTRPV3 portals, which connect the membrane environment to the central cavity of the pore. The inhibition is likely achieved due to a barrier for ion conductance that dyclonine molecules form by sticking out into the channel pore. The binding of small-molecule inhibitors to the TRPV3 pore portal was shown for the first time, but it was originally proposed to be the mechanism of action for local anesthetics on voltage-gated sodium channels [87]. Cryo-EM structure of wild-type hTRPV3 in complex with trpvicin shows that the inhibitor stabilizes the channel in the closed conformation by binding to the residues in S4, S5, and S5 segments. The complex with a gain-of-function G573S-TRPV3 mutant demonstrates that trpvicin accesses additional binding sites inside the central cavity of the constitutively open channel and remodels the channel symmetry [84].

## 5. TRPV3 Ligands in the Research of Channel’s Physiology

### 5.1. Chemo- and Thermosensation

TRPV3 is abundantly expressed in the skin (keratinocytes), especially in the hair follicles and the basal layer of the epidermis in mammals [33,88,89,90,91,92]. TRPV3 is also expressed in human sensory neurons, spinal motor neurons, and peripheral nerves [7,8,93], but it is still unclear whether TRPV3 is expressed in rat and mouse sensory neurons [6,55]. In addition, TRPV3 expression is present in the gastrointestinal tract, including the epithelium of the oral cavity, tongue, larynx, distal colon, jejunum, and ileum [33,51,94,95,96,97].

Sensory neurons operate in their microenvironment, which is determined by complex interactions of neurons with non-neuronal cells, including immune cells, neuronal accessory cells, fibroblasts, adipocytes, and keratinocytes (the phenomenon reviewed in [98]). In this case, the transduction of sensory (as well as pain) signals to sensory nerve endings is realized through some chemical messengers. TRPV3 in humans and mice contributes to the perception of warm ambient temperature (more than 33 °C) by epidermal keratinocytes and transduces signals through ATP-dependent pathway to the free nerve endings of local sensory neurons [6,7,8,99,100,101,102]. TRPV3 activation by warm temperature elevates the level of intracellular Ca^2+^ in keratinocytes. This evokes ATP release which, in turn, activates ATP receptors in the termini of dorsal root ganglion neurons [99,101,102]. However, as shown on knockout mice, TRPV3 contribution to the perception of a warm temperature is rather limited, and largely relies on the cooperation with TRPV1-dependent and other mechanisms [103,104].

In the early research of TRPV3 agonists, special consideration has been given to skin sensitizers, i.e., agents that modulate skin sensitivity to heat in humans. For example, camphor activated primary cultured mouse keratinocytes but not sensory neurons, and this activity was abolished in TRPV3-null mice [100]. The components of plant-derived essential oils, like thymol, carvacrol, and eugenol, also activated TRPV3 channels of oral and nasal epithelium, where they likely act as chemesthetic receptors (along with TRPV1, TRPV3, TRPA1, TRPM5, and TRPM8) and modulate or initiate signals of the nervous system [33]. Sensory compounds that evoke sensations of heat or cold were shown to have promiscuous action on thermoTRP channels. Menthol, which is known for its cooling effect via cold-sensing TRPM8, also activates TRPV3 in high millimolar concentrations [56] and induces TRPA1 activation in low micromolar concentrations and a reversible channel block in higher concentrations [105]. Similarly, camphor activated TRPV1 (better than V3) but inhibited TRPA1, and cinnamaldehyde activated TRPV3 and TRPA1 but inhibited TRPM8 [56]. Citral, a component of lemongrass, showed complex modulation of sensory TRP channels (TRPV1-V4, TRPA1, TRPM8). Regarding TRPV3, citral produced activation followed by rapid desensitization [55].

### 5.2. Hair Growth

TRPV3 is abundantly expressed in the epithelium of human follicles during anagen and catagen stages and in the outer root sheath keratinocytes, but not in dermal papilla fibroblasts and dermal fibroblasts. This expression is not regulated by cycling signals of hair follicles, as the expression level is almost the same at different stages of hair development [88,91]. TRPV3 is required for the normal hair growth and development. Underexpression of TRPV3 in mice causes wavy hair coat and curly whiskers [100]. Conversely, the gain-of-function mutant TRPV3 leads to hair loss due to impaired hair follicle regeneration because of the stimulation of late keratinocyte differentiation and apoptosis. The gain-of-function mutant TRPV3 is assumed to downregulate transcription regulators FOXN1, MSX2, DLX3, and GATA3 that are critical for regulating follicular keratinocyte differentiation [88,106,107,108,109]. TGF-α/EGFR signaling axis, which is critical for the hair follicle and hair shaft development, was also associated with TRPV3 functioning [16,110]. Activation of EGFR in keratinocytes is accompanied by the activation of TRPV3 channels, which stimulates TGF-α release [16]. TRPV3 knockout mice have phenotypes with curly whiskers and wavy hair that are very close to phenotypes of mice with loss-of-function TGF-α and EGFR gene mutations [100,111].

Chemical activation of TRPV3 by eugenol, carvacrol, and 2-APB inhibits human hair shaft elongation, keratinocyte proliferation in a time- and dose-dependent manner, and also induces the catagen stage of the hair follicle cycle and keratinocyte apoptosis [90,91]. The specific inhibition of TRPV3 by forsythoside B or short-hairpin RNA reversed the cell death induced by carvacrol-mediated TRPV3 activation in human outer root sheath cells. Forsythoside B also significantly reversed hair growth inhibition induced by carvacrol [90].

### 5.3. Brain Functions

TRPV3 protein was not detected in the mouse brain, although low levels of TRPV3 mRNA transcripts were detected [112,113]. TRPV3 mRNA is expressed in the cerebellum and hypothalamus of rats [114,115] and the human brain [7]. Brown et al. reported that the deletion of TRPV3 channels can influence synaptic plasticity in hippocampus. Preparations from TRPV3-knockout mice, along with TRPV1-, but not TRPV4-knockouts, lacked long-term depression at excitatory synapses on hippocampal CA1 interneurons and showed reduced long-term potentiation at CA3-CA1 pyramidal cell synapses [116].

Incensole acetate, a potent agonist of TRPV3, was used as a probe to reveal some psychoactive functions of TRPV3 [47]. Intraperitoneal injection of incensole acetate caused multiple emotional and behavioral effects in mice, which were accompanied by changes in c-Fos activation in several areas in the brain. The association of TRPV3 activation with anxiolytic and antidepressant-like effects of the compound was confirmed on TRPV3-null mice where the effects were abolished [47]. These observations are valuable evidence of the diversity of TRPV3 functions in the body and in the CNS.

Additional anti-inflammatory and protective potential of incensole acetate was shown in mice with cerebral ischemic injury. The effects were associated with the inhibition of NF-kB and were partially attenuated in TRPV3-deficient mice [117].

TRPV3 overexpression was reported to increase neural excitability and exacerbate ischemic injury. Specific inhibition of TRPV3 by forsythoside B decreased neural excitability and attenuated cerebral I/R injury in mice. Thus, blocking overactive TRPV3 may provide a potential therapy to promote recovery from ischemic brain stroke [118].

## 6. TRPV3 Ligands in the Research and Correction of Channel’s Pathophysiology

### 6.1. Pain

The physiological contribution of TRPV3 to pain signals is still not fully understood. TRPV3 is expressed in sensory neurons such as the dorsal root ganglia (DRG) and in the trigeminal ganglia (TG) in humans and nonhuman primates [8]. In many human tissues, TRPV1, functionally significant in neuropathic pain conditions, and TRPV3, are co-expressed and can form heteromeric channels, which was not observed in mice [20]. Perhaps this underlies the key regulatory differences between the two species. TRPV3 is co-expressed with TRPV1 in a discrete subpopulation of vagal afferent neurons and may contribute to vagal afferent signaling in a similar capacity as TRPV1 [119]. It was shown that, along with TRPV1, TRPV3 expression is increased in peripheral nerves after injury, and the expression of both channels is markedly reduced in skin with diabetic neuropathy [93]. At the genetic level, potentially pathogenic variants of TRPV3 genes were identified in patients with small fiber neuropathy [120], migraine [121], and erythromelalgia [122]. Sensory hypoinnervation of the epidermis and, as a result, a deficit in sensations of acute mechanical pain, cold, and itching is observed in gain-of-function mutations of TRPV3, e.g., G573S [5]. In a behavioral pain model, mice with a mutant hyperactive TRPV3 (G573S) channel showed an increased mechanical threshold for pinpoint mechanical pain and a reduced response to acute pain compared to control littermates [123].

In the electrophysiological study in vivo, McGaraughty et al. showed that synthetic TRPV3 antagonists modulated the activity of key classes of pain pathway neurons by limiting the transmission of pathological nociceptive signals through receptors in the periphery and in the brain. The blockade of supraspinal TRPV3 receptors reduced both evoked and unprovoked pain, so the therapeutic potential of TRPV3 antagonists can be modified if they access the channels in the brain [124].

Under the action of TRPV3 agonists (eugenol, 2-APB) or heat stimulation, skin keratinocytes secrete pro-inflammatory mediators (IL-1α, PGE2, TGF-α, ATP) [33,101,125]. ATP released by keratinocytes upon TRPV3 activation interacts with the purinoergic receptors of sensory neurons, thereby causing pain and neurogenic inflammation [126,127,128]. Interestingly, several pro-inflammatory agents (bradykinin, PGE2, histamine, ATP) could increase the sensitivity of TRPV3 to warm temperatures, leading to an autocatalytic TRPV3-mediated increase in skin inflammation and associated symptoms, such as thermal hyperalgesia [23,129,130].

The analgesic activity of TRPV3 ligands has been shown in a number of animal model tests (Table 3).

TRPV3 agonist farnesyl pyrophosphate (FPP) reduced TRPV3 heat threshold, resulting in enhanced behavioral sensitivity to noxious heat in mice. The administration of FPP also elicited pain in carrageenan-treated inflamed mice, and the effects were prevented by pre-treatment with ruthenium red or by TRPV3 knockdown. It was noteworthy that FPP induced nocifensive behavior only under inflamed conditions [45]. An inhibitor isopentenyl pyrophosphate (IPP) suppressed acute nociception induced by cinnamaldehyde, and TRPV3-mediated FPP-induced pain in an inflamed paw. In addition, IPP reduced thermal and mechanical hyperalgesia and mechanical allodynia induced by complete Freund’s adjuvant, but in TRPV3-null mice, only the heat analgesia by IPP was significantly blunted [46]. The administration of an inhibitor 17(R)-resolvin D1 reduced FPP-induced pain and reversed the thermal, but not the mechanical, hypersensitivity elicited by the inflammatory response, and the effect was blunted in mice with a local knockdown of TRPV3 [58].

In another study, a TRPV3 inhibitor citrusinine II alleviated nociceptive behavior in mice in a number of pain models, showing a weak analgesic activity [61]. A potent and selective TRPV3 antagonist 74a from AbbVie Inc. demonstrated a favorable preclinical profile in two different models of neuropathic pain as compared to the reference drug, gabapentin, and evoked antinociception in a reserpine model of central pain, but effective doses (30–100 mg/kg) were too high for further development [81].

### 6.2. Itch, Skin Inflammation, and Dermatitis

TRPV1 and TRPA1 are key participants in pain and inflammation [131,132] but their link to itch has also become evident in recent years [133]. Apparently, the channels act downstream of a number of G-protein coupled receptors that sense pruritogens, or via cytokine or toll-like receptors, or directly sensing electrophilic irritants (e.g., TRPA1). Expression of TRPV1 and TRPA1 is increased in skin inflammatory and allergic conditions. TRPV1-knockout animals also have a reduced histamine-induced itch reaction, and histamine-sensitive small-diameter DRG-neurons respond to capsaicin, a TRPV1 agonist [133]. Recent research indicates that TRPV3 also plays a critical role in inflammatory skin diseases [9,125,134]. Mutant animals exhibit excessive scratching, indicating a role for TRPV3 in itch [89,135]. TRPV3-knockout animals also show a significant reduction of histamine-induced itch [59]. TRPV3 expression was increased in skin biopsies from patients with post-burn pruritic scars [136]. The link between TRPV3 and protease-activated receptor 2 (PAR2) was found in conditions of atopic dermatitis [128].

Carvacrol exerted non-monotonic effects on human keratinocyte culture HaCaT promoting proliferation at concentrations up to 100 μM but decreasing cell viability at 300 μM. The proliferation effect was abolished with the suppression of TRPV3 expression, and, thus, it is likely dependent on TRPV3-mediated Ca^2+^ influx [137]. In vivo, intradermal injection of carvacrol induced pruritus in mice that was mediated by TRPV3 and TRPA1. Carvacrol activates both channels with similar potency, but the activation of TRPA1 is followed by a rapid desensitization [33,138]. The pattern of the scratching behavior was two-phase, where the initial phase supposedly represented a joint contribution by TRPA1 and TRPV3. The following sustained phase was mainly mediated by TRPV3, which was evidenced on TRPV3-knockout mice in which this phase was reduced to approximately 90% [138]. A carvacrol-based animal model has been used for the evaluation of the antipruritic effects of TRPV3 inhibitors (examples below).

Among the drugs that are already in medical use, dyclonine, applied topically for pain and itch relief in mucous membranes and skin, was reported to be an effective TRPV3 inhibitor. Dyclonine significantly suppressed TRPV3-mediated scratching behavior induced by carvacrol in comparison with a knockout control group [78].

TRPV3 inhibitors from medicinal plants show positive therapeutic effects against pruritus, dermatitis, and skin inflammation in murine models, which establishes them as promising experimental therapeutics for cutaneous diseases (Table 4). Forsythoside B effectively reduced cytotoxicity by carvacrol in vitro, suppressed acute itch induced by pruritogens (carvacrol or histamine), and attenuated scratching behavior in a model of dry skin by acetone-ether-water (chronic itch) in mice [75]. A closely related compound verbascoside produced beneficial effects against carvacrol-induced pruritus and skin inflammation [76]. Coumarin osthole significantly attenuated mouse scratching behaviors induced by either acetone-ether-water or histamine, but not in TRPV3-deficient mice [59]. Similarly, citrusinine II attenuated acute and chronic itch in mice induced by acetone-ether-water or histamine, but not chloroquine [61]. Isochlorogenic acid A (IAA) and isochlorogenic acid B (IAB) were assessed in vivo in a model of pruritus induced by chronic application of carvacrol. Transdermal application of the compounds attenuated both ear swelling and scratching behavior [62]. Administration of scutellarein in mice suppressed atopic dermatitis induced by 2,4-dinitrofluorobenzene and pruritus induced by carvacrol and alleviated epidermal hyperplasia and skin inflammatory response, but showed no benefit in TRPV3-knockout mice in the carvacrol model. In vitro, scutellarein suppressed carvacrol-induced proliferation of HaCat keratinocytes, expression and release of chemotactic factors, and chemotaxis of peripheral blood mononuclear cells [63].

A synthetic compound, trpvicin, which potently inhibited TRPV3 currents in vitro, attenuated scratching behavior in mice treated with PAR2 agonist SLIGRL upon intradermal injection. Systemic administration of trpvicin in high doses (100 mg/kg) also reduced scratching bouts and ear swelling in mice treated with calcipotriol [84].

### 6.3. Olmsted Syndrome

TRPV3 is robustly linked to a rare hereditary disease named Olmsted syndrome (OS). Currently, three genes, mutations in which lead to the development of the disease, are known: recessive X-linked mutation of *mbtps2* [139,140], autosomal dominant mutation of *perp* [141,142], and autosomal dominant or recessive mutation of *trpv3* [143,144]. The classic characteristic features of Olmsted syndrome are bilateral mutilating palmoplantar keratoderma and periorificial keratotic plaques (around the mouth, nose, eyes, external auditory canal, umbilicus, and anogenital region). The disease is accompanied by severe pain, itching, and burning in the area of hyperkeratotic foci, a pathology of hair and nails, corneal lesions, leukokeratosis, and an increased tendency to recurrent infections [145].

TRPV3-determined Olmsted syndrome is associated with more than 20 amino acid substitutions that lead to the channel gain-of-function (Figure 3). The substitutions in G573, G568, and L673 are the most common, but mutations can also occur in W692, W521, Q580, M672 and some other positions. Most mutations are located in the S4-S5 linker or the C-terminal part of the protein [144,145]. Expression of OS mutants (G573S, G573C, G573A and W692G) causes disruption of vesicular transport, which leads to a decreased surface localization of these mutants themselves and a closely related TRPV1 channel as well. Expression of the mutants also causes reduced cell adhesion, altered distribution of endoplasmic reticulum and Golgi bodies, and fewer lysosomes. [146]. TRPV3 with substitutions at G568C and G568D also caused a decrease in cell size and severe defects in the number of lysosomes and their distribution. Both mutant proteins were unable to mobilize Ca^2+^ from intracellular stores, especially when cytosolic Ca^2+^ was chelated and/or in the absence of extracellular Ca^2+^. They also disrupted pH maintenance in lysosomes, most likely acting as Ca^2+^ leak channels [143].

The symptoms of Olmsted syndrome are thought to be related to the TRPV3/EGFR signaling pathway. Treatment of patients with Olmsted syndrome (gain-of function mutation Gly573Cys) by the EGFR inhibitor erlotinib led to positive results a rapid significant improvement in the symptoms of palmoplantar keratoderma (PPK), including pain and itch. Increased EGFR phosphorylation in the skin of OS patients supports the theory that TRPV3 activation stimulates EGFR signaling and that EGFR may serve as a missing link between TRPV3 mutations and activation of the mTOR pathway [149].

Current treatments of Olmsted syndrome include systemic retinoids, corticosteroids, and other anti-inflammatory and pain-relieving drugs, which, however, provide only a temporary symptomatic relief [145]. The use of TRPV3 antagonists could represent a novel targeted strategy for relieving Olmsted syndrome symptoms. A local anaesthetic dyclonine was tested against TRPV3 with gain-of-function mutations G573S and G573C in vitro, and showed an effective inhibition of the leak currents and cell death caused by the overactive mutants [78]. Forsythoside B also attenuated cell death in HEK293 or HaCaT cells heterologously expressing the constitutively open TRPV3-G573S mutant in a dose-dependent manner [75]. Compound trpvicin inhibited the currents of hTRPV3-G568V and mTRPV3-G568V heterologously expressed in HEK293T cells. Trpvicin was also tested on heterozygous *trpv3^+/G568V^* knock-in mice that showed symptoms similar to those of patients with Olmsted syndrome, especially hair loss. Mice topically treated on their backs with a cream containing 1% trpvicin showed substantially longer hair shafts and less hair shedding throughout the period of treatment [84].

### 6.4. Wound Healing and Barrier Functions

TRPV3 is included in the skin barrier functions and participates in wound healing. Wound healing in *trpv3*-knockout mice was much slower than in the wild type [150]. Activation of TRPV3 by farnesyl pyrophosphate (FPP), an endogenous ligand, improved skin wound healing and resulted in an efficient bacterial clearance. Endogenous expression of TRPV3 protein was found in the macrophages, where it was present in lysosomes and the nucleolus. In hyperthermic shock conditions, the presence of TRPV3 in nucleolus correlated with lysosomal function and stress level [151]. According to the other data, TRPV3 mRNA expression was practically undetectable in unstimulated or LPS-stimulated mouse macrophages [152]. Other data indicated that TRPV3 regulates nitric oxide synthesis in the skin, thus promoting wound healing in vivo [153]. A synthetic activator KS0365 accelerated keratinocyte migration that paralleled strong TRPV3-mediated calcium responses in migrating front cells and in leading edges, which suggests the role of TRPV3 in re-epithelialization upon skin wounding [54].

## 7. Conclusions

Industrial pharma actively contributes to the design and study of TRPV3 antagonists, which is manifested in a growing number of related patents. However, to date, such compounds hardly ever entered clinical trials [9,154,155]. The rare example is compound GRC15300 developed by Sanofi-Aventis/Glenmark that reached but failed phase II trial for patients with peripheral neuropathy [155]. For the review of patents regarding the topic, one can address [9,154]. The problem with the study of TRPV3 inhibitors as analgesic agents is that, despite a significant number of patents, few molecules and pre-clinical trials can be found in peer-reviewed literature. This deters the evaluation of this therapeutic strategy per se. Apparently, the mechanism of analgesia is complex and may involve supraspinal nociceptive pathways and non-neuronal microenvironment of peripheral nerves, i.e., keratinocytes. Keratinocytes initiate, transduce, and modulate sensory and pain signals with an active participation of TRPV3. It is noteworthy that TRPV3 is not the only channel that can mediate this keratinocyte function. Mechanical stimulation of keratinocytes, via the mechanosensitive ion channel Piezo1, can lead to the release of neuroactive mediators that activate sensory nerve terminals. Voltage-gated sodium channels Na_V_1.5, Na_V_1.6, and Na_V_1.7 on keratinocytes may be involved in several pain conditions [98].

Concerning the therapy of dermatitis and pruritus, TRPV3 inhibitors, indeed, possess a significant potential. Good efficacy was shown for such routes of administration as intradermal and transdermal in a form of creams/oils. Pathological channel isoforms also respond to TRPV3-targeted pharmacological treatment, although pre-clinical studies are lacking. The only pre-clinical experiments on heterozygous *trpv3^+/^^G568^^V^* knock-in mice were made for trpvicin and showed positive effects on hair shafts.

The progress in deciphering cryo-EM structures of TRPV3, structural mechanisms of action of different ligands, and establishing binding sites opens possibilities for further rational design of selective and high-affinity ligands. New potent, selective, and non-toxic molecules may present lead compounds for the development of target-selective drugs and new tools for the further study of the TRPV3 function in native tissues.

## Figures and Tables

**Figure 1 ijms-24-08601-f001:**
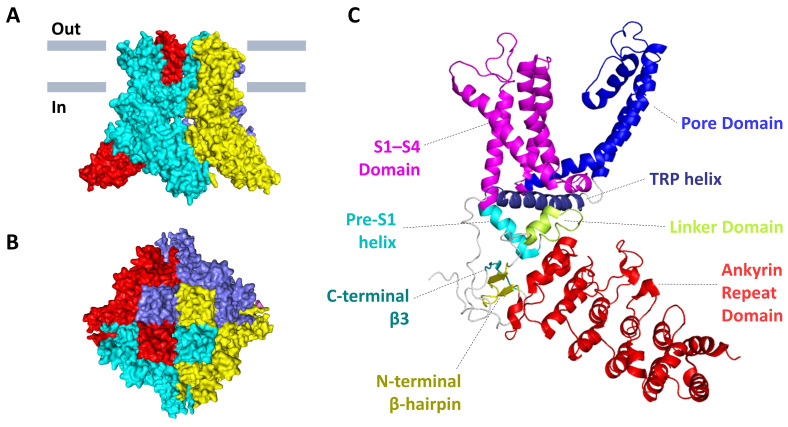
The 3D-structure of mouse TRPV3 in a closed apo-state (PDB: 6DVW). (**A**) Side and (**B**) top views of the tetramer protein. Individual subunits are colored in red, cyan, yellow, and slate. (**C**) The structural organization of a single subunit with domain mapping.

**Figure 2 ijms-24-08601-f002:**
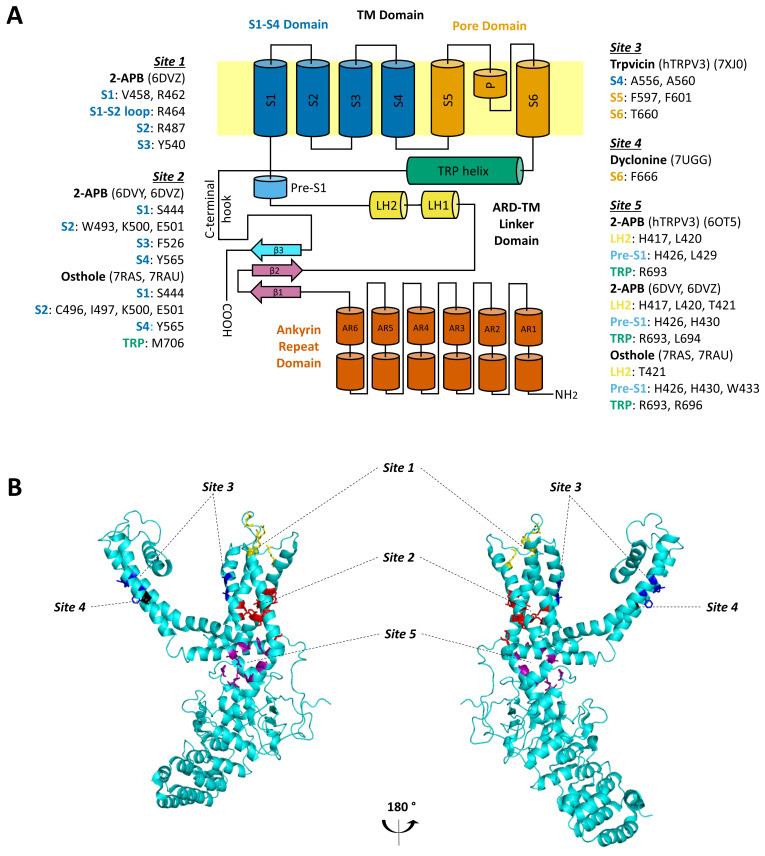
TRPV3 binding sites of exogenous ligands from cryo-EM data. (**A**) The architecture of a single TRPV3 subunit is presented as a topological diagram. Protein data bank (PDB) accession IDs are given in parentheses. The data shown refer to mouse TRPV3 unless noted otherwise. AR1–6—ankyrin repeats 1–6, ARD-TM—ankyrin-repeat domain- transmembrane domain, hTRPV3—human TRPV3, β1–3—β-strands 1–3, LH1–2—linker helix 1–2, P—pore helix, pre-S1—pre-S1-helix, S1–S6—transmembrane segments 1–6, TM—transmembrane. (**B**) The 3D view of the binding sites on a single TRPV3 subunit mapped by different colors.

**Figure 3 ijms-24-08601-f003:**
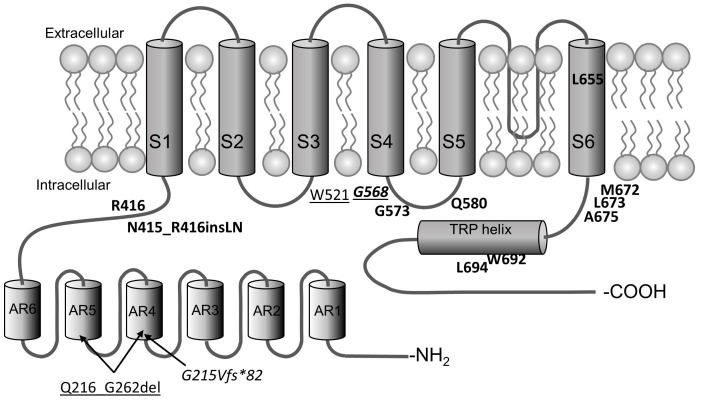
Schematic arrangement of mutations detected for Olmsted syndrome. Dominant mutations are indicated in bold, recessive mutations are underlined, and semidominant mutations are indicated in italics [143,145,147,148].

**Table 3 ijms-24-08601-t003:** Analgesic efficacies of TRPV3 inhibitors.

Experimental Model	Tested Compound	Effective Dose/Concentration, Route of Administration	Ref.
Formalin-induced pain	Citrusinine-II	2.9, 14.3 mg/kg, i.p. *	[61]
Acetic acid-induced writhing	Citrusinine-II	2.9, 14.3 mg/kg, i.p.	[61]
Hot plate test	Citrusinine-II	2.9, 14.3 mg/kg, i.p.	[61]
Tail flick test	Citrusinine-II	1.5, 2.9, 14.3 mg/kg, i.p.	[61]
Complete Freund’s adjuvant-induced inflammation, heat pain threshold	17(R)-resolvin D1	30 µM in 20 µL, intraplantar	[58]
	Isopentenyl pyrophosphate	1 mM, i.d.**	[46]
Complete Freund’s adjuvant-induced inflammation, FPP-induced pain	17(R)-resolvin D1	30 µM in 20 µL, intraplantar	[58]
Carrageenan-induced inflammation, FPP-induced pain	Isopentenyl pyrophosphate	1 mM, i.d.	[46]
Chronic constriction injury	74a	30, 100 mg/kg, p.o. ***	[81]
Sciatic nerve ligation	74a	30, 100 mg/kg, p.o.	[81]
Reserpine-induced central pain	74a	10, 30, 100 mg/kg, p.o.	[81]

* i.p.—intraperitoneal, ** i.d.—intradermal, *** p.o.—per oral.

**Table 4 ijms-24-08601-t004:** Antipruritic efficacies of TRPV3 inhibitors.

Experimental Model	Tested Compound	Effective Dose/Concentration, Route of Administration	Ref.
Histamine-induced pruritus	Forsythoside B	0.3, 3, 30 µM, i.d. *	[75]
	Osthole	300 nM, i.d.	[59]
	Citrusinine-II	10 µM, i.d., t.d. **	[61]
Carvacrol-induced pruritus	Forsythoside B	30, 300 µM, i.d.	[75]
	Verbascoside	300 µM, i.d.	[76]
	Isochlorogenic acid A	1 mM, t.d.	[62]
	Isochlorogenic acid B	1 mM, t.d.	[62]
	Scutellarein	0.2, 0.5 mg/kg/day, s.c. ***	[63]
	Dyclonine	10, 50 µM, i.d.	[78]
Acetone-ether-water-induced pruritus	Forsythoside B	3, 30 µM, i.d.	[75]
	Osthole	30, 300 nM, i.d.	[59]
	Citrusinine-II	5, 10 µM, i.d.	[61]
2,4-dinitrofluorobenzene-induced dermatitis and pruritus	Scutellarein	0.2, 0.5 mg/kg/day, s.c.	[63]
SLIGRL-induced pruritus	Trpvicin	10, 100 µM, i.d.	[84]
Calcipotriol-induced pruritus	Trpvicin	100 mg/kg/day, p.o. ****	[84]

* i.d.—intradermal, ** t.d.—transdermal, *** s.c.—subcutaneous, **** p.o.—per oral, once daily starting from 5 days before topical treatment by calcipotriol.

## Data Availability

Not applicable.

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
