# Peer review of "TRPV3 Ion Channel: From Gene to Pharmacology"

_ijms, 2023, doi:10.3390/ijms24108601_

Round 1
Reviewer 1 Report
This review article is an exhaustive account of TRPV3 ion channel and well written. I have few comments about the article -
1 - Page 2 - line 81 to 93 - these two paragraphs are about protein and are more suitable for section 2.2.
2 - While the structure of TRPV3 is well described, a figure showing the structural elements is more helpful. one figure showing the tetramer with different color for each subunit and another showing the structural element in one subunit. The elements can be colored differently.
3 - Similarly, in figure 1, showing various ligands/drug in the context of structure will be more informative. the existing figure can also be included specially for ligands where the structural information are not available.
At few places the writing can be better. Few examples are -
1 - line 44 - there is a number -->> there are number.....
2 - line 45 - lack of potent and highly selective pharmacology.
3 - line 132 - the b3 bonds with is giving the impression of covalent bond. please specify and clarify.
Author Response
We thank the Reviewer for the attention to our manuscript and positive feedback. Below is our point-by-point response.
1 - Page 2 - line 81 to 93 - these two paragraphs are about protein and are more suitable for section 2.2.
In these paragraphs, we aimed to put emphasis on the evolutionary adaptations of the channel in different organisms. We suppose it would still be more suitable for the section 2.1, because section 2.2 focuses on the murine and human proteins. To avoid confusion of readers, we suggest renaming the section 2.1 as “Gene and evolution”.
2 - While the structure of TRPV3 is well described, a figure showing the structural elements is more helpful. one figure showing the tetramer with different color for each subunit and another showing the structural element in one subunit. The elements can be colored differently.
We are grateful for the recommendation and add a new figure in the section 2.2 that shows the structure of the tetramer and the domains of the single subunit (fig. 1).
3 - Similarly, in figure 1, showing various ligands/drug in the context of structure will be more informative. the existing figure can also be included specially for ligands where the structural information are not available.
We add a panel to this figure (now fig. 2) that shows a 3D view of the binding sites on a single subunit.
At few places the writing can be better. Few examples are -
1 - line 44 - there is a number -->> there are number.....
Corrected as suggested.
2 - line 45 - lack of potent and highly selective pharmacology.
Corrected to “lack of potent and highly selective pharmacological tools”.
3 - line 132 - the b3 bonds with is giving the impression of covalent bond. please specify and clarify.
According to (doi: 10.1038/s41594-018-0108-7), in mouse TRPV3, the beta3-strand of the C-terminus associates with the beta-hairpin (made of 2 antiparallel strands) of the N-terminus to form a 3-stranded antiparallel beta-sheet. This structure tethers cytoplasmic N- and C-termini within one subunit. The nature of bonding here is apparently classical for the beta-sheet formation: hydrogen bonds. This structural motif was earlier found in TRPV1, a closely related ion channel (doi.org/10.1038/nature12822). Besides, authors note this was earlier found in G-protein-coupled inwardly rectifying potassium channels (“the β-sheet structure tethers cytoplasmic N- and C-terminal domains together within the same subunit, reminiscent of G-protein-coupled inwardly rectifying potassium channels, in which N- and C termini interact through a short parallel β-sheet”, doi.org/10.1038/nature12822).
So, in our manuscript, we suggest the following correction: “The β3 connects to the β-hairpin in the linker domain via hydrogen bonds to form a three-stranded β-sheet that, together with the C-terminal hook, participates in intersubunit interactions with the ARD that stabilize the elements of the intracellular skirt together”.
Reviewer 2 Report
This submission is a well-written and interesting manuscript which brings much recent data regarding TRPV3 pharmacology.
I noticed only some minor points which need correction:
1. Line 45 - please replace"selective pharmacology" with "selective ligands/pharmacological tools" or similar. Pharmacology is a field of science, so this statement does not make sense.
2. As this is a review paper, please avoid "personal" reference as in line 78 - "this allows us".
3. The title of the section 3 needs to be re-written - just simply "Ligands of TRPV3" or "Available ligands of TRPV3"
4. In my opinion it is not necessary to refer to compounds like 7c, 8c (line 269), 74a (line 279, Table 3), etc. Alternatively, maybe their chemical structures should be shown?
5. Abbreviations are not explained at their first use (Table 4 - DNFB, SLIGRL, Table 3 -CFA).
Author Response
We are grateful to the Review for the time and effort devoted to our manuscript and for positive remarks. Below is our detailed response.
- Line 45 - please replace "selective pharmacology" with "selective ligands/pharmacological tools" or similar. Pharmacology is a field of science, so this statement does not make sense.
We are sorry for this slang expression. Corrected to “pharmacological tools”.
- As this is a review paper, please avoid "personal" reference as in line 78 - "this allows us".
Corrected to “it assumes that…”.
- The title of the section 3 needs to be re-written - just simply "Ligands of TRPV3" or "Available ligands of TRPV3"
Corrected to “Ligands of TRPV3”.
- In my opinion it is not necessary to refer to compounds like 7c, 8c (line 269), 74a (line 279, Table 3), etc. Alternatively, maybe their chemical structures should be shown?
We agree that this naming is awkward. However, these are designer compounds that never received trivial names in the original articles and are referred to with these number-letter designations. So, addressing original articles would be necessary for a potential reader. We also would not like to show chemical structures as they would overburden the table and would add little information in the context.
- Abbreviations are not explained at their first use (Table 4 - DNFB, SLIGRL, Table 3 -CFA).
DNFB, CFA are explained in the text. SLIGRL, agonist of PAR-2 receptor, is a hexapeptide named after constituting amino acids (S-serine, L-leucine, etc.), so a proper name is used.